# Recent Progress of Copper-Based Nanomaterials in Tumor-Targeted Photothermal Therapy/Photodynamic Therapy

**DOI:** 10.3390/pharmaceutics15092293

**Published:** 2023-09-07

**Authors:** Xiqian Zhuo, Zhongshan Liu, Reyida Aishajiang, Tiejun Wang, Duo Yu

**Affiliations:** Department of Radiotherapy, The Second Affiliated Hospital of Jilin University, Changchun 130062, China; zhuoxq23@mails.jlu.edu.cn (X.Z.); liuzhongshan@jlu.edu.cn (Z.L.); rydas21@mails.jlu.edu.cn (R.A.)

**Keywords:** nanomaterials (NMs), photodynamic therapy (PDT), photothermal therapy (PTT), copper (Cu), Cu-based NMs, reactive oxygen species (ROS), glutathione (GSH)

## Abstract

Nanotechnology, an emerging and promising therapeutic tool, may improve the effectiveness of phototherapy (PT) in antitumor therapy because of the development of nanomaterials (NMs) with light-absorbing properties. The tumor-targeted PTs, such as photothermal therapy (PTT) and photodynamic therapy (PDT), transform light energy into heat and produce reactive oxygen species (ROS) that accumulate at the tumor site. The increase in ROS levels induces oxidative stress (OS) during carcinogenesis and disease development. Because of the localized surface plasmon resonance (LSPR) feature of copper (Cu), a vital trace element in the human body, Cu-based NMs can exhibit good near-infrared (NIR) absorption and excellent photothermal properties. In the tumor microenvironment (TME), Cu^2+^ combines with H_2_O_2_ to produce O_2_ that is reduced to Cu^1+^ by glutathione (GSH), causing a Fenton-like reaction that reduces tumor hypoxia and simultaneously generates ROS to eliminate tumor cells in conjunction with PTT/PDT. Compared with other therapeutic modalities, PTT/PDT can precisely target tumor location to kill tumor cells. Moreover, multiple treatment modalities can be combined with PTT/PDT to treat a tumor using Cu-based NMs. Herein, we reviewed and briefly summarized the mechanisms of actions of tumor-targeted PTT/PDT and the role of Cu, generated from Cu-based NMs, in PTs. Furthermore, we described the Cu-based NMs used in PTT/PDT applications.

## 1. Introduction

Cancer is a complicated disease characterized by various genetic flaws. The increased incidence of malignant tumors and their predisposition for metastasis are major threats to human health, which lead to higher mortality rates [1,2]. The traditional treatment modalities for tumors, including surgery, radiotherapy, and chemotherapy, can engender various adverse effects, including radioactive damage, toxic side effects of chemotherapeutic drugs, or chemotherapy-induced multidrug resistance, which considerably limits the therapeutic effectiveness of these modalities [3,4,5,6]. The emerging treatment modalities for tumors, such as thermotherapy, immunotherapy, and gene therapy, have provided greater hope for patients [7,8]. However, their prolonged use has been reported to severely impair the immune system, even leading to organ malfunction [9,10]. Discovering a tumor treatment modality with low toxicity to healthy tissues and improved precision presents an imperative and formidable challenge.

Recently, phototherapies (PTs), such as photothermal therapy (PTT) and photodynamic therapy (PDT), are being widely used as effective antitumor therapeutic strategies [10]. In PTT and PDT, photothermal agents (PAs) and photosensitizers (PSs) work as exogenous energy converters or absorbers in the organ affected by the tumor [11]. This has enabled the conversion of visible or laser light energy into thermal energy to induce apoptosis or necrosis of tumor cells at high temperatures, with or without concurrent reactive oxygen species (ROS) generation [12,13,14,15]. PT has specific advantages over other heat-based tumor treatment techniques, including thermotherapy and microwave therapy. Its high precision enables PT to target tumors with the least possible damage to adjacent tissues and organs [16]. Additionally, PT is an excellent method for managing tumors because of its controllability and low toxicity profile. Compared with either PTT or PDT acting alone, the synergistic approach combining PTT and PDT can hasten tumor cell death and increase the therapeutic efficacy [17]. The key components of PTT and PDT are their PAs and PSs, majorly including nanomaterials (NMs) at present, which differ from conventional macromolecules. NMs exhibit excellent characteristics, including high Brunauer–Emmett–Teller (BET), electrical conductivity, spectrum shifts after light absorption, fluorescence properties, and potential for degradation [18]. Therefore, they have been steadily incorporated into cancer research, advancing the field of tumor investigations [19,20,21]. Regarding medical therapeutics, materials of nanoscale dimensions (1–100 nm diameter) have been employed [22]. These NMs have versatile applications in drug transportation, enabling controlled release mechanisms, increasing permeability, traversing biological barriers, and improving overall biocompatibility [23,24,25]. Owing to their unique size and traits, NMs can remarkably change various therapeutic processes and considerably improve treatment efficacy [22]. Metal ion-based NMs, including those of gold (Au), silver (Ag), copper (Cu), and other NMs, exhibit distinctive optical properties, notably the phenomenon of localized surface plasmon resonance (LSPR) [26,27]. LSPR notably strengthens the electric field in the immediate proximity of metal ion-based NMs, based on their capacity to strongly absorb photon energy. Owing to an extensive range of light-induced effects and intermolecular interactions produced, they are used in tumor-targeted PTT/PDT. However, the high cost of Au- or Ag-based NMs and the non-degradability of some PAs/PSs in vivo limits their application in clinical therapy [28]. Cu, a readily available metal, has unique bioactivities that can be used for eliminating tumor cells by regulating various types of cell death [29]. Furthermore, Cu can undergo a Fenton-like reaction to catalyze high ROS content from excess intracellular hydrogen peroxide (H_2_O_2_), which can induce bacterial and tumor cell death [30]. The functions and therapeutic involvement of Cu are shown in Figure 1. Al Kayal et al. created electrospun polyurethane membranes covered with Cu nanoparticles (NPs) and ran an antimicrobial test on *Escherichia coli* and discovered that over half of the bacteria were destroyed, showing its potent bactericidal effects. Additionally, SARS-CoVer-2 was resistant to the antiviral activity by nearly 90% [31]. Cu-based NMs have been widely employed in PTT and PDT in recent years because of their advantageous properties, including strong near-infrared (NIR) absorption and photothermal capabilities [32,33], high BET [34,35], and use in tumor imaging. [36] Furthermore, Cu-based NMs offer considerable advantages in tumor therapy owing to their simple synthesis procedures and relatively high production yields under mild low reaction conditions [37]. Zhao et al. synthesized CCeT NMs (Ce: Chlorin e6, T: TPP-COOH) with Cu_2−x_ Sulfur (S) as the core, which exhibited a photothermal conversion efficiency (PCE) of 10.6% under laser irradiation. At very low concentrations, the temperature increased by 5.3 °C in 10 min and continued to rise with increasing concentration [38], demonstrating the therapeutic advantages of PTT/PDT and that it can be substantially increased using Cu-based NMs. 

Herein, we reviewed Cu-based NMs and explored the role of Cu in synergizing PTT/PDT for tumor cell death. This review also provides a comprehensive overview of the Cu-based NMs utilized in PTT/PDT.

## 2. Principles and Drawbacks of PTT/PDT in Antitumor Treatment and the Benefits of Cu-Based NMs

### 2.1. Principles and Drawbacks of PTT/PDT in Antitumor Treatment

PTT can be divided into traditional PTT (≥45 °C) and mild PTT (MPTT, 42–45 °C) [39]. It is an antitumor therapy where PAs convert laser light energy into thermal energy under NIR irradiation, raising the temperature of the surrounding region and inducing apoptosis or necrosis in the tumor cells [40]. In PDT, PSs generate ROS in the presence of oxygen molecules upon or following photoirradiating of the tumor tissue, resulting in oxidative damage-induced tumor cell death [41]. PDT is often classified as type I or type II, where type I produce O_2_^−^, hydroxyl radicals (·OH), and H_2_O_2_, and type II produce ^1^O_2_ [42,43]. Although PTT and PDT may both cause tumor cell death by phototoxicity and producing ROS, they still have notable drawbacks in actual use [44].

First, the practical usage of PTT/PDT is generally constrained by PAs/PSs because of the low tissue penetration of light, which leads to poor tumor cell death, tumor resistance to therapy, and possible excruciating pain during treatment [45,46]. Additionally, with the rise in local temperature around the tumor, adjacent normal tissues and cells may also suffer damage. Finally, ROS production is dependent on the oxygen present around the tissue, but the low oxygen content in the tumor microenvironment (TME) severely restricts the effectiveness of PDT [47]. To resolve these problems, NMs are incorporated into PTT and PDT; Cu-based NMs can act as carriers or modified PAs or PSs to fix the limitations of conventional PAs or PSs [43,48]. For example, Zhang et al. produced biodegradable cancer cell membrane-coated mesoporous Cu/manganese (Mn) silicate nanospheres (m@CMSNs) to target the aggregation of homozygous cancer cells via adhesion molecules on the surface of cancer cell membranes [49,50]. Under 635 nm laser irradiation, m@CMSNs were targeted to aggregate 2.85-fold more than CMSNs at the tumor site and degraded by glutathione (GSH) in the TME. Cu^1+^ functions as a very effective PSs, producing huge levels of ^1^O_2_ at the tumor site, killing tumor cells and reducing tumor cell viability to 20% [51]. Thus, the deficiencies of conventional PAs/PSs, which are especially addressed in detail in the next section, can be significantly improved by Cu-based NMs.

### 2.2. Biological Properties of Cu and Benefits of Cu-Based NMs

Cu is the third most common essential trace element after zinc and iron [52]. Despite being in low concentrations, Cu is widely distributed in biological tissues and plays roles in many critical processes, including cellular respiration, energy metabolism, and ROS removal [53,54]. Both Cu deficit and surplus conditions can cause serious diseases [52,55]. Several studies have shown increased Cu levels (2–3 times higher) in patients with tumors compared with that in healthy individuals [56,57]. Hence, the strict regulation of Cu levels in cells and tissues is essential.

Owing to the special biological characteristics of Cu, Cu-based NMs provide many benefits in antitumor therapy: (1) Cu-based NMs have favorable NIR absorption and excellent photothermal performance because of the d-d transition of Cu^2+^, the unpairing of 3d electrons and paramagnetic of Cu^2+^ [58]. They also demonstrate the capability of magnetic resonance imaging (MRI), which has been widely used in PTT and photoacoustic imaging (PAI) of tumors [59,60]. Because Cu^2+^ in small Cu sulfide (S) NPs (s-Cu_2-x_S NPs) have unpaired electrons, they are magnetic and can be employed as possible contrast agents for T1-weighted MRI [61]. Chen et al. synthesized ultrasmall structured u-Cu_2-x_S (Cu^2+^) nanodots and found an increase in the photoaccoustic signal in mice, especially at the tumor site. In particular, the PA signals could sustain markedly higher intensities for extended periods [62]. Weitz et al. synthesized CuO NPs; a quantitative estimation of MRI model experiments compared to water showed a slope of 54.1 (95% CI: 30.93–77.26) for the percent change in signal per mg Cu/mL [63]. (2) Cu-based NMs are biodegradable and have a high BET for loading various chemotherapeutic drugs for tumor chemodynamic therapy (CDT) [64]. He et al. synthesized HKUST-1, a Cu-based metal–organic framework (MOF) that could be hydrolyzed by GSH in the TME, and the released sorafenib and meloxicam (Mel) effectively limited tumor cell growth and operated as a CDT [65]. Pang et al. synthesized CuTz-1-O_2_@F127 NPs, which could carry dioxygen (O_2_) owing to their high BET and be degraded by TME. The released O_2_ reduced tumor hypoxia while increasing the therapeutic effects of PDT. Additionally, feces and urine accounted for the excretion of approximately 90% of NPs, suggesting that the NPs would be rapidly cleared post treatment [66]. (3) Cu-based NMs are considered effective Fenton-like catalysts. Cu^2+^ can help with anticancer PDT for CDT because at neutral pH Cu^2+^ interacts >100 times faster with H_2_O_2_ compared with Fe^2+^, accelerating the production of large quantities of ·OH and O_2_ from excess intracellular H_2_O_2_ [67]. Li et al. synthesized CuFeSe_2_–LOD@Lipo-CM nanocatalysts (LOD: lactate oxidase, Lipo-CM: liposome containing glioblastoma cell membrane proteins). In the TME, lactate oxidase increased H_2_O_2_ levels and the released Cu^1+^ reacted with H_2_O_2_ to produce ·OH, leading to CDT. Furthermore, modest warming of the tumor site via NIR irradiation can increase the antitumor therapeutic effects of CDT [65]. Li et al. synthesized GSH-degrading CFT@IP6@BSANPs (CFT: Cu iron tellurite, IP6: inositol hexaphosphate, BSA: bovine serum albumin) that effectively degraded overexpressed GSH in the TME. The simultaneous release of Cu^1+^ loaded within the NMs, which acted as chemotherapeutic agents, induced tumor cell deaths [68]. (4) Cu-based NMs can produce high ROS quantities when exposed to light; therefore, they are frequently exploited as PAs/PSs for PTT/PDT of malignancies. Qu et al. synthesized hollow CuS nanocubes that increased temperature in a concentration-dependent manner under 808 nm laser irradiation, with a PCE of 30.3%; additionally, increased ROS generation was detected which induced tumor cell deaths, acting as an antitumor therapy [69]. Therefore, Cu-based NMs with exceptional characteristics have been successfully created to be used in the combined treatment of malignancies to give Cu-based NMs extra functionalities. PTT/PDT and Cu-based NMs can both be used to treat cancer in different ways and Cu-based NMs can compensate for some of the drawbacks of PTT/PDT. Thus, Cu may synergize with PTT/PDT to induce tumor cell death in various ways to achieve antitumor effects.

## 3. Cu-Based NMs Synergizes PTT/PDT-Induced Tumor Cell Death

### 3.1. Cu-Based NMs Synergizes PTT/PDT-Induced Tumor Cell Necrosis

PTT/PDT can eliminate tumor cells via multiple pathways. When exposed to exogenous light, PSs/PAs attached to tumor cell membranes and lysosomes may directly lead to tumor cell necrosis [70]. Cu has a photothermal ability and a high NIR absorption capacity; therefore, Cu-containing PSs/PAs localized in the tumor cell membrane interact with the surrounding oxygen molecules during NIR irradiation, leading to an elevated energy level and the formation of ^1^O_2_ through oxygen energy transfer. Then, ^1^O_2_ oxidizes unsaturated fatty acid constituents of the membrane to malondialdehyde, damaging the integrity of the membrane which causes rapid utilization of intracellular adenosine triphosphate (ATP), inducing necrosis [71,72]. Proteins and lipids, which make up the majority of the tumor cell membrane, are photosensitive and their chemical transformation may swiftly destroy cells, even when light is administered at low temperatures. Necrotic tumor cells can attract antigen-presenting cells and deliver tumor-associated antigens to naive T cells [73], which leads to the activation of cytotoxic T cells present near the tumor tissue and damage-associated molecular patterns (DAMPs) from dying tumor cells, inducing the immune response [74]. After light irradiation and tumor cell death, phosphatidylserine may present to macrophages, inducing immunosuppressive cytokine generation, which inhibit the growth of antigen-presenting dendritic cells (DCs) and reduces inflammation [75]. Additionally, DCs are further activated by DAMPs, which stimulate the immune system and initiate antitumor immunological responses, increasing their capacity to uptake tumor-associated antigenic epitopes [76]. Jin et al. synthesized LPS–CuS NPs (LPS: lipopolysaccharide), induced tumor ablation via laser irradiation, and found increased interleukin (IL)-6, IL-12p40 and tumor necrosis factor-α mRNA levels in tumor-draining lymph nodes. The mRNA levels of T-helper-1 cell transcription factors interferon and T-bet expressed in T cells were also increased, which increased tumor antigen-specific immune response by promoting DC activation [77]. Jiang et al. synthesized AuNBP@CuS (NBP: nanobipyramid) NMs with a core/shell design and found that NIR irradiation increased extracellular ATP, calreticulin (CRT), and high mobility group box-1 (HMGB-1) levels. Both of them are immunological signals that, within a specified concentration range, can effectively increase the expression of CD80 and CD86 signals in DCs and promote DCs maturation [78]. Studies have shown that PTT and Cu-based NMs worked synergistically to create DAMPs from dying cells and activate immunological responses. The illustration of Cu-based NMs synergizes PTT/PDT-induced tumor cell necrosis is shown in Figure 2.

### 3.2. Cu-Based NMs Synergizes PTT/PDT-Induced Apoptosis

Cu is a crucial trace element for the organism and several studies have shown its harmful effects. In a study, a Cu-based treatment of apoptotic factors increased B-cell leukemia/lymphoma-2-associated X-protein (Bax), B-cell leukemia/lymphoma-2-associated agonist of cell death protein (Bad), cytochrome c (Cyt c), and caspases 3 and 9 levels [79]. By binding to apoptosis-activating caspase 9, the Cu induced an increase in Bax level, causing the release of Cyt c and mitochondrial apoptosis-inducing factor 1 (Apaf-1) from the mitochondria into the cytoplasm, where it activated caspase 3 to cause apoptosis. Simultaneously, ROS caused DNA breakage in PC12 cells and activated the caspase cascade response [79]. Additionally, Cu-induced ROS increased lipid peroxidation and decreased GSH levels, making cells more vulnerable to oxidative stress-induced harm [80,81]. Hosseini et al. reported that Cu^2+^ interacted with respiratory complexes (I, II, and IV) and promoted ROS generation in mouse hepatocytes in vitro [82]. Mitochondria is also an important target for PAs/PSs. Hilf showed that PSs enter tumor cells and are rapidly taken up by mitochondria, triggering mitochondria-mediated apoptosis in response to light and greatly reducing the mitochondrial ATP concentration. PDT produces ROS, which oxidizes the phospholipids in the mitochondrial membrane by involving oxygen molecules in energy or electron transport. Depolarization and a decrease in the mitochondrial membrane potential cause mitochondrial expansion, which damages the outer mitochondrial membrane [83]. Both PTT and PDT were capable of producing ROS when exposed to light irradiation. The application and breakdown of Cu-based NMs increased the amount of Cu ions in tumor cells, decreasing the amount of GSH and promoting ROS accumulation [84]. Mitochondrial damage prompted by high ROS leads to Cyt c releasing into the cytoplasm and upregulation of the Bax gene. The increase in Cu ions also resulted in an elevated production of Bax, Cyt C, and caspase3/9, which, when combined with PTT/PDT, led tumor cells to undergo an apoptotic cascade reaction [79]. Mithun et al. synthesized biotin–Cu@AuNP and discovered that when laser irradiated, NPs accumulated considerably in the mitochondria of A549 cells. Moreover, the caspase3/7 activity notably increased, suggesting that NPs triggered apoptosis via the caspase-activation pathway [85]. In summary, Cu can induce the production of pro-apoptotic factors and ROS through direct or indirect effects on mitochondria, PTT/PDT can cause ROS damage to mitochondria through phototoxicity, and Cu-based NMs synergizes PTT/PDT-induced apoptosis, increasing the death of tumor cells and resulting in anti-tumor therapy. The illustration of Cu-based NMs synergizes PTT/PDT-induced apoptosis is shown in Figure 3.

### 3.3. Cu-Based NMs Synergizes PTT/PDT-Induced Autophagy

Many studies have demonstrated that Cu ions can trigger and inhibit autophagic processes via different pathways [86,87]. Polishchuk et al. analyzed gene enrichment using the ATP7B mouse model and found that an insufficient level of ATP7B restricted Cu efflux and increased intracellular Cu concentration [88]. The autophagy marker MAP1LC3 (also known as LC3) level was greatly increased, indicating that autophagy was activated while also interfering with the inactivation of the mammalian target of rapamycin (mTOR)-dependent signaling on the lysosomal membrane [89,90]. Autophagy is still promoted by mTOR deactivation because it can stimulate dephosphorylation and transcription factor EB (TFEB) translocation to the nucleus [88,91]. The unc-51-like autophagy activating kinase (ULK)1/2 is a known downstream target of mTOR. Donita C et al. discovered that ULK1 kinase activity was increased in Cu-sufficient cells and decreased in Cu-deficient cells, indicating that the Cu level directly affected ULK1/2 activity [92,93]. The Cu transporter 1 (Ctr1) deletion or ULK1 mutation inhibited Cu binding, which increased intracellular Cu levels and boosted ULK1 kinase activity, producing an autophagy complex to trigger autophagy [94]. Furthermore, Ctr1 deletion, considered pertinent to the high Cu-content-induced autophagy, inhibited tumor development in the Kirsten rat sarcoma virus G12D-driven lung cancers [95]. Qin et al. reported that copper sulfate (CuSO_4_) increased mitochondrial ROS formation (mtROS) and activated autophagy in RAW264.7 cells via the protein kinase B/adenosine monophosphate-activated protein kinase/mTOR pathway. Additionally, rapamycin-stimulated autophagy decreased apoptosis, whereas autophagy-related 5 knockdown to promote autophagy increased CuSO_4_-induced apoptosis [96]. Li et al. reported HYF127c/Cu, a completely novel Cu combination. The transcriptome sequencing revealed that some autophagy genes, including the MAP1LC3B gene, were considerably upregulated in HYF127c/Cu-treated HeLa cells and that HYF127c/Cu was reported to activate autophagy in HeLa cells [97]. Studies have shown that ROS produced by PDT can activate and result in a long-lasting increase in autophagy. Whether PTT/PDT-induced survival autophagy is primarily protective or death autophagic is still undetermined [98]. Numerous studies have suggested that thermal stress induces pro-survival autophagy in tumor cells, resulting in treatment resistance in PTT/PDT [99]. Cu-based NMs are capable of carrying autophagy inhibitors. Cu ions can improve anti-tumor efficacy by inducing pre-survival autophagy, inhibiting the heat-induced pro-survival autophagy produced by PTT/PDT and generating ROS for CDT treatment through a Fenton-like reaction, combining PTT/PDT and CDT to achieve the improvement of antitumor efficacy [29]. For example, Wen et al. synthesized Cu palladium alloy tetrapod NPs (TNP-1) with good photothermal characteristics that also activated autophagy due to an increased Cu^2+^ release. In the 4T1 and MCF7/multidrug-resistant breast cancer models, the combination of TNP-1 with the autophagy inhibitors CQ or 3-MA significantly enhanced the anticancer efficacy of TNP-1-mediated PTT [100]. Although the exact mechanism of PTT/PDT-induced autophagy is yet to be known, Cu-based NMs may trigger autophagy in PTT/PDT to induce antitumor effects. The illustration of Cu-based NMs synergizes PTT/PDT-induced autophagy is shown in Figure 4.

### 3.4. Cu-Based NMs Synergizes PTT/PDT-Induced Pyroptosis

Cu^2+^ can be reduced by GSH to Cu^1+^ in the TME that overexpress GSH. Cu^1+/2+^ and PTT/PDT may together activate the generation of ROS under UV irradiation, resulting in cancer cell pyroptosis [101]. In a study, CuSO_4_ markedly expedited cellular pyroptosis via the NLRP3/caspase-1/GSDMD pathway. Furthermore, the pyroptosis-associated genes IL-1 and NLRP3 exhibited an increased expression in hepatocytes, which promotes inflammation and causes the extravasation of immune cells [102,103,104]. The inflammasome caspase-1 is essential for pyroptosis, a highly inflammatory form of programmed cell death [105,106]. In cancer cells, increasing the intracellular ROS levels activates the nucleotide-binding oligomerized structural domain-like receptor protein 3 (NLRP3) inflammasome and causes pyroptosis [107]. Wang et al.′s synthesized GOx@Cu MOFs produced excessive ·OH radicals, utilizing the internal biological cascade reaction of glucose oxidase (GOx) and the Fenton-like reaction of Cu ions, resulting in tumor cell pyroptosis [108]. In the work by Yin et al., it was discovered that higher Cu^2+^ concentrations and longer exposure times decreased the protein content of the cAMP/PKA/CREB pathway, as well as the potential of the mitochondrial membrane and the amount of GSH-Px (Glutathione peroxidase) in MN9D cells. Concurrently, enhanced ROS generation, as well as the expression of Nrf2, NQO1, HSP-70, and other proteins, resulted in the creation of inflammasome and mediated the overexpression of GSDMD proteins, resulting in pyroptosis in MN9D cells [109]. PTT/PDT therapy is a promising strategy for non-invasively inducing pyroptosis in cancer cells [105]. PDT effectively initiates pyroptosis via inflammasome activation and the production of ROS; PTT can accelerate this process, making it the ideal complement to enhance photoimmunotherapy′s efficacy [110]. Tang et al. designed the aggregation-induced dimeric photosensitizer D1 (Figure 5b). This compound targets tumor cell membranes and, when used with PTT/PDT, proves highly effective in ROS production and, subsequently, pyroptosis induction [102]. The surge in intracellular ROS contributes to the pyroptosis instigated by Cu ions. At present, the specific research mechanism of Cu-induced cell pyroptosis in PTT/PDT is still unclear, but Cu and PTT/PDT operate through distinct mechanisms to induce tumor cell death. Their combined anti-tumor approach should offer synergistic effects, exemplified by the principle “1 + 1 > 2.” The illustration of Cu-based NMs synergizes PTT/PDT-induced pyroptosis is shown in Figure 5a.

### 3.5. Cu-Based NMs Synergizes PTT/PDT-Induced Cuproptosis

The phenomenon of cuproptosis emerges when an excessive concentration of Cu accumulates within a cell. This buildup promotes the enrichment of lipoylated dihydrolipoamide S-acetyltransferase (DLAT). Subsequently, DLAT disruption affects the tricarboxylic acid cycle (TCA) during mitochondrial respiration, inducing proteotoxic stress and leading to cell death [111]. Xie et al. developed a Cu^2+^-doped nanohybrid gel integrated with a PTT/PDT/CDT therapeutic combination [112]. The NIR-responsive capability of Cu^2+^ harmonizes with Au in the gel, optimizing PTT/PDT tumor treatment [113]. Simultaneously, Cu^2+^ mitigates the challenge of tumor hypoxia, enhancing the therapeutic efficacy of PDT. This effect also results in the degradation of Cu ions due to the excessive accumulation of Cu^2+^ in TMEs with a low pH. Additionally, Cu^2+^ interacts with GSH, transforming into Cu^1+^ and GSH oxidized (GSSG), enabling CDT to produce more harmful ·OH radicals [112,114]. Furthermore, in vitro data revealed that the mortality rate of activated cancer cells was greater than 81.4%, while tumor suppression in vivo was as high as 85.7%, demonstrating cuproptosis-enhanced PDT/PTT/CDT against malignant tumors. Zhu et al. synthesized Cu-doped polymetallic oxalate nanoclusters (Cu-POM), enhancing their uptake in cells via MPTT, leading to intracellular Cu-POM accumulation. Excessive Cu interrupts the initiation of the TCA cycle and promotes intracellular peroxide formation, culminating in cuproptosis [115]. Therefore, cuproptosis offers a novel therapeutic approach for tumor treatment by regulating intracellular Cu levels in tumor cells. Moreover, Cu plays a pivotal role in PTT/PDT antitumor strategies. The illustration of Cu-based NMs synergizes PTT/PDT-induced cuproptosis is shown in Figure 6.

In conclusion, Cu and PTT/PDT may both be employed in various ways in anti-tumor therapy (Figure 7). In treating tumors, they are complimentary to one another. Compared to a single therapy method, this synergistic approach can produce better anti-tumor benefits.

## 4. Application of Different Kinds of Cu-Based NMs in PTT/PDT

### 4.1. Application of Copper Oxides in PTT and PDT

Over the past few decades, various Cu-based NMs have emerged as alternative means to amplify the effects of PTT/PDT [119]. Studies indicate that CuO-NPs can hinder pancreatic tumor growth, notably by targeting tumor stem cells [120]. These NPs can induce mitochondrial dysfunction, attributed to the ability of Cu^2+^ to generate ROS through the Fenton-like reaction and to enhance PCE through electronic transitions between C-2p and Cu-3d [121]. Ma et al. developed CuO@CNSs-DOX (DOX: Doxorubicin) nanoplatforms, elevating the PCE from 6.7% to 10.14%. These platforms achieve anti-tumor effects through the drug action of DOX and the generation of ·OH radicals by Cu^2+^ (Figure 8b) [122,123]. The same group synthesized multifunctional MoS_2_-CuO@BSA/R837 (MCBR, BSA: Bovine Serum Albumin, R837: Imiquimod, a toll-like receptor 7 (TLR7) agonist) nanoflowers. Under 808 nm laser exposure, these nanoflowers exhibited a PCE of 24.6% and a marked increase in ·OH production. Additionally, the inclusion of R837 increased the expression of calreticulin (CRT), triggering immunogenic cell death (ICD), effectively neutralizing primary tumors, and inhibiting metastatic tumors (Figure 8a) [124]. Taking advantage of Cu_2_O’s high refractive index above 600 nm, Yu et al. produced core/shell structured Cu@Cu_2_O@polymer NPs. Under a 660 nm laser exposure and an equivalent Cu concentration, compared to Cu@polymer NPs, the temperature increase for Cu@Cu_2_O@polymer NPs was about 4.2 °C. This resulted in a temperature rise of at least 23 °C in adjacent tissues, translating to a 7-fold surge in the IC_50_, relative to prior research [125,126]. These findings underscore that Cu@Cu_2_O@polymer NPs possess sufficient phototoxicity to exterminate tumor cells. Furthermore, the increased concentration of endogenous H_2_O_2_ produced by the LPS endotoxin accelerated the release of Cu ions and ROS, facilitating the eradication of cancer cells (Figure 8c) [33,125].

### 4.2. Application of Cu_x_S_y_ in PTT and PDT

Semiconductor NMs Cu_x_S_y_ have gained attraction in catalysis and sensing sectors due to their optical and electrical properties. Owing to their affordability, minimal cytotoxicity, and high photostability, Cu_x_S_y_ NPs hold promise as Pas [127,128]. Tian et al. synthesized hydrophilic, plate-like Cu_9_S_5_ nanocrystals. With a 980 nm laser, the PCE of Cu_9_S_5_ reached 25.7%. Interestingly, under identical conditions, this PCE surpassed that of the synthesized Au nanocrystals, which was approximately 23.7%. These nanocrystals eradicated tumor cells in vivo swiftly (in less than 10 min), establishing Cu_9_S_5_ nanocrystals as efficient PAs. When hormonal mice underwent subsequent radiation following injection, notable necrosis and shrinkage of tumor cells in the mass region were observed [129,130]. Addressing the challenge that non-targeted ultrasmall metallic NPs face in securing prolonged half-life and satisfactory tumor-site aggregation as PTT agents, Li et al. synthesized CuS NPs and modified them with cetuximab (Ab) to yield CuS-Ab NPs. On exposure to a 1064 nm laser for 10 min, temperatures increase rapidly from 23 °C to 58 °C and are consistently held at 58 °C. Additionally, Ab modification rendered CuS-Ab NPs superior in tumor-targeting and anti-angiogenic capacities, ensuring effective tumor-site accumulation without concomitant damage to other tissues and organs [131]. Cationic polymers, such as PEI, can achieve drug delivery by intensifying the electrostatic adsorption of cations on the cell membrane [132]. Studies have shown that NMs modified with PEI and PSs can be instrumental in the PDT of bladder tumors [133]. Based on this, Mu et al. prepared HRP@CPC@HA NPs (HRP: horseradish peroxidase, CPC: CuS-PEI-Ce6, HA: hyaluronic acid). These particles incorporated HRP within a hollow CuS nanocage [134]. Subjected to 808 nm laser radiation, their PCE was around 34.91%. Concurrently, the surface-bound Ce6 was activated, leading to a large amount of ^1^O_2_ generation, the levels of which increased alongside temperature [133]. Additionally, HRP catalyzed the breakdown of intracellular H_2_O_2_ to oxygen, addressing the oxygen deficit in tumor cells and facilitating simultaneous multimodal tumor eradication in severely hypoxic TME [135]. Due to the increased concentration of endogenous H_2_S in tumor cells, Wu et al. synthesize monolocking NPs (MLNPs) [136]. When illuminated with 808 nm radiation, these MLNPs combined with H_2_S and produced ultra-small CuS nanodots. This interaction elevated tumor temperatures, triggering apoptosis in tumor cells. Mel is released from the NPs which causes the COX-2 enzyme to become inactive, amplifying the PTT action and inhibiting inflammation induced by PTT damage. Notably, these NPs were excreted renally (Figure 8e,f) [137]. The development of nanomedicine has witnessed diverse manifestations of Cu_x_S_y_. Hollow structured Cu_x_S_y_ has particularly piqued interest in oncologic therapy, attributed to its drug-loading ability and high PCE and PAI capability [138,139], yet many extant synthesis methods for CuS-based NMs are intricate, time-consuming, and require special reaction equipment. Consequently, there is an evident need for facile, cost-effective methodologies that employ gentle reaction conditions to fabricate Cu_x_S_y_ NMs.

### 4.3. Application of Copper Selenides and Copper Telluride in PTT and PDT

Copper selenides, notably Cu_2-X_Se NMs, present promising applications in imaging and PTT for cancer treatment, attributed to their biocompatibility, excellent NIR light absorption, and increased PCE (Figure 8g) [140,141].

Du et al. synthesized CuSe/NC-DOX-DNA NPs (NC: nitrogen-doped carbon) via an environmentally friendly and simple method. Under 808 nm laser exposure, these NPs displayed a PCE of 32.9%, surpassing that of Au nanorods which had a PCE of 22.0% [142]. Moreover, the synergistic action of photocatalysis combined with the Fenton-like reaction of CuSe/NC markedly augmented ROS generation [143]. Concurrently, the encapsulated DOX functioned as a chemotherapeutic agent, realizing a threefold enhanced PTT/CDT/PCT tumor treatment method [144]. It is noteworthy that Cu_2-X_Se has been identified as an effective PA [141]. In another study, He et al. developed ICG@Cu_2-X_ Se-ZIF-8 (ICG: Indocyanine Green, ZIF-8: a metal-organic framework). After 808 nm laser exposure, this compound exhibited a PCE of 15.5%. In the TME, ZIF-8 breaks down and releases Cu^1+^ and Cu^2+^, which leads to a Fenton-like reaction, controls the amount of GSH, and produces ROS (Figure 8d) [145]. Furthermore, selenium has the ability to regulate selenoprotein, allowing it to prevent the production of osteoclasts and tumor cells, resulting in the synergistic PTT/CDT prevention of malignant bone metastasis [146,147]. Moreover, CuSe is an ideal PS and has been demonstrated to biodegrade. Selenium is released during this process and has been shown to lower the risk of liver cancer, lung cancer, and prostate cancer [148,149]. Pun et al. fabricated COF-CuSe NMs (COF: covalent-organic framework); under 808 nm laser irradiation, the PCE was 26.34% after injection in mice. A large quantity of ^1^O_2_ was generated under both 650 nm and 808 nm illumination. Consequently, almost all of the HeLa cells died under laser irradiation. The PTT and PDT anti-tumor treatment strategies showed a synergistic potential (Figure 8h) [150]. At present, CuSe NMs have been poorly investigated for PTT/PDT and the development of multifunctional CuSe NM is still required to improve anti-tumor therapy for PTT/PDT.

**Figure 8 pharmaceutics-15-02293-f008:**
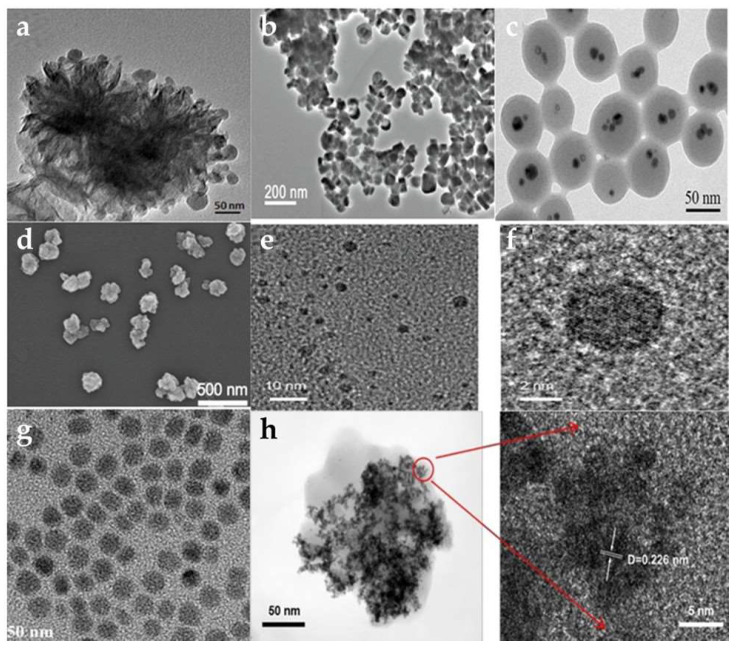
Transmission electron micrographs (TEM) of different kinds of Cu-based NMs. (**a**) [124] 2021 Elsevier, (**b**) [123] 2011 American Chemical Society, (**c**) [33] 2018 American Chemical Society, (**d**) [145] 2022 John Wiley and Sons, (**e**,**f**) [137] 2021 John Wiley and Sons, (**g**) [140] 2013 John Wiley and Sons, and (**h**) [150] 2022 American Chemical Society.

Similar to CuSe, CuTe is considered a novel PAs candidate. Li et al. created CuTe NPs with a plasma peak at 900 nm. Under 830 nm laser irradiation, the mortality of 3T3 embryonic fibroblasts increased [151]. However, it was discovered that some cells were already dead before laser irradiation. It has been demonstrated that CuTe NPs are cytotoxic and PAs. Under 1064 nm laser irradiation, bio Cu_2−X_Te nanosheets synthesized by Li et al. had a PCE of 48.6%. When Cu^1+^ and Te are released, the nanosheets can produce ·OH and inhibit the GPx and TrxR enzymes for CDT, significantly inhibiting the proliferation of MCF-7 cells [152]. Shen et al. synthesized CM CTNPs@OVA NMs (CM: melanoma B16-OVA membrane, OVA: ovalbumin) with solid CuTe NPs. Under laser irradiation, there was a significant increase in temperature and production of ROS. In addition, B16-OVA cells produced an abundance of ATP and HMGB-1, which effectively stimulated the immune system and enhanced the anti-tumor treatment [153]. CuTe NPs anti-tumor therapy research in PTT/PDT has received less attention and requires further development and investigation.

### 4.4. Application of Cu-Based Nanocomposites in PTT and PDT

Most treatments consisting of a single therapy have a negligible impact on tumor treatment. However, a synergistic combination of multiple therapeutic modalities can improve the efficacy of treating malignant tumors [154]. The same applies to NMs. In cancer treatment, simple Cu-based NMs such as CuS may still be lacking. Therefore, it is necessary to load various materials, drugs, or fluorescents onto Cu-based NMs to create a composite material capable of fluorescence, tumor targeting, and therapy in a single NM.

As a result of the discovery of cuproptosis, Cu-based NMs may induce tumor cell death by modulating the concentration of Cu in tumor cells, offering a novel anti-tumor therapeutic modality [111]. Pan et al. produced GOx@[Cu(tz)] NPs. Under 808 nm laser irradiation, NPs entering tumor cells produced H_2_O_2_ and ·OH. In the meantime, under the influence of GSH, GOx hydrolyzed and consumed glucose, generating a large amount of H_2_O_2_ and OH that produced a ROS-adding effect [155]. Due to the depletion of glucose and GSH, the NPs bind to lipoylated mitochondrial enzymes, resulting in the aggregation of lipoylated DLAT, which induces cuproptosis and effectively inhibits tumor growth (92.4% inhibition rate) [156]. The synthesis and development of NMs that combine multiple antitumor therapeutic modalities is urgently needed. Xia et al. synthesized metal-organic skeleton nanosheets Cu-TCPP(Al)-Pt-FA (TCPP: Tetrakis (4-carboxyphenyl) porphyrin, FA: folic acid) with surface modification by platinum NPs (Pt NPs) and FA in order to solve the problem of poor oxygenation in tumor tissues. Compared to 25.2% tumor cell survival in vitro without laser irradiation, laser irradiation at 638 nm reduced tumor cell survival to 20.7%. Since Cu^2+^ can react with GSH via a Fenton-like reaction, it depletes intracellular GSH and increases ROS levels [157]. In the meantime, Pt NPs have catalysis activity comparable to a catalase-like reaction that can continuously convert intracellular H_2_O_2_ to O_2_ in order to alleviate hypoxic TME and enhance the therapeutic effect of PDT [158]. ROS concentration was increased synergistically by these two modalities, which stimulated antigen-presenting cells to activate systemic anti-tumor immune responses and increased the infiltration of cytotoxic T lymphocytes (CTLs) at the tumor site for synergistic immune anti-tumor therapy [159,160]. Xu et al. constructed d-Cu-LDH/ICG NPs (LDH: lactatedehydrogenase) in order to alleviate the tumor hypoxia problem and avoid the poor therapeutic effect caused by tumor hypoxia during PDT treatment [161]. Under 808 nm laser irradiation, the PCE was 88.7% and the production of ^1^O_2_ increased as the temperature rose. In the meantime, the rising temperature led to the dissolution of the NMs and the reduction in Cu^2+^ to Cu^1+^ by GSH, both of which can consume excessive H_2_O_2_ and generate OH in tumor cells via a Fenton-like reaction, resulting in CDT and modulation of the TME [162]. In the meantime, it was demonstrated that hormonal mice had significantly less tumor growth. Hematoxylin and eosin (H&E) staining of major organs revealed no obvious inflammation or damage and demonstrated that the effective anti-tumor agent exhibited no significant systemic toxicity. Yang et al. created NSCuCy NPs containing Cu^1+^ as the core [163]. Cu^1+^ undergoes a Fenton-like reaction with O_2_^−^ to produce ·OH [164]. The fluorescence intensity of the HPF was used to detect the ROS concentration; it was discovered that the fluorescence intensity of the HPF increased rapidly in the presence of NSCuCy NPs under 660 nm irradiation. At pH 5.5, the emission intensity of HPF increased nearly 350-fold after 10 min of irradiation, demonstrating a higher ·OH production efficiency than that at pH 7.4 (180-fold) and targeting accumulation in tumor tissues to achieve complete tumor ablation [165]. Kang et al. developed Au@MSN-Cu/PEG/DSF NPs (Au@MSN: mesoporous silica-coated Au nanorods, DSF: disulfiram). The PCE under 808 nm laser irradiation is 56.32%. With an increase in temperature, the Cu-doped SiO_2_ framework begins to biodegrade. During the conversion of Cu^2+^ to Cu^1+^, releasing DSF can chelate with Cu^2+^ to produce highly cytotoxic bis (diethyldithiocarbamate) Cu (CuET) [166,167]. Cu^1+^ binds additionally to mitochondrial protein aggregates during the TCA cycle, inducing cuproptosis in tumor cells [111]. Photothermal therapy′s synergistic effect resulted in an 80.1% tumor inhibition rate, effectively killing tumor cells and inhibiting tumor growth [111,168].

Overall, multifunctional nanocomposite materials that integrate imaging, diagnosis, and therapy have shown significant improvements in tumor treatment compared to single-material therapy. These NMs combine PTT/PDT with drug delivery systems, immunotherapy, and chemotherapy, reducing normal medication dosage and adverse reactions while achieving effective anti-tumor treatment in the short term (Figure 9). However, long-term experimental data on NMs are still needed to verify their long-term toxicity, biosafety, and therapeutic effects to ensure they meet desired goals. Further investigations are required for a comprehensive understanding.

### 4.5. Application of Cu-MOF in PTT and PDT

Metal-organic frameworks (MOFs) are a novel type of NMs that combine metal ions or clusters with multisite organic ligands. These MOFs have shown great potential in anti-tumor therapy due to their inherent biodegradability, high porosity, structural diversity, and high drug-loading capacity [170,171,172,173].

For instance, Wu et al. synthesized ultrathin Cu-TCPP MOF nanosheets containing both Cu^1+^ and Cu^2+^ which enabled imaging, photothermal conversion, and anti-tumor therapy [36,174]. Under 808 nm laser irradiation, the nanosheets showed a PCE of 36.8% and a significant amount of ^1^O_2_ was generated in tumor cells. Cu^2+^ has unpaired 3d electrons and paramagnetic also allowed the nanosheets to be used for T1-weighted MRI. This multifunctional NMOF design holds promise for tumor PTT [59]. Tang et al. designed two metal-organic materials, MOF-1 and MOF-2, based on this premise. MOF-1 is an aluminum (AL)-MOF that does not contain Cu^2+^, whereas MOF-2 is an AL-Cu mixed-metal MOF[CuL-[AlOH]_2_n with Cu^2+^ at its core. Comparing the two MOFs under 650 nm laser irradiation revealed that MOF-2 has a porous physical structure, which makes the Cu^2+^+ in MOF-2 significantly adsorb intracellular GSH and results in a decrease in GSH and an increase in ROS concentration, which enhances the efficacy of PDT [175]. Similar to the chemotherapeutic drug camptothecin (CPT), MOF-2 was able to eradicate tumor cells in an in vivo experiment. The results of a simultaneous examination of major organ tissue slices revealed little toxicity in vivo [176,177]. This MOF structure, which simultaneously decreases intracellular GSH levels and increases intracellular ROS levels, may provide not only a new strategy for intracellular GSH adsorption but also a new method for enhancing PDT.

Due to the higher cavity structure of emerging Cu-NMOFs, they can be loaded with more PSs or PAs to improve PCE and increase the diffusion range of ROS. Their porous structure also allows for chemotherapeutic drug delivery and synergistic effects with other therapies to enhance anti-tumor efficacy. However, the synthesis process of MOF PSs or PAs can be complex and might exhibit batch-to-batch variations, hindering large-scale NMOF preparation. Moreover, the presence of various metal ions necessitates further investigation into the long-term safety, biocompatibility, pharmacokinetics, and immunoreactivity of NMOFs in mammals through systematic animal studies.

## 5. Conclusions and Future Perspectives

One of the most significant challenges in cancer treatment today is achieving precise targeting. PTT and PDT have demonstrated promising results for early surface cancers. Moreover, combining various PAs and PSs has significantly improved the therapeutic efficacy of PTT/PDT for deep tumors. Recently, Cu-based NMs have emerged as promising candidates due to their excellent NIR absorption, paramagnetic characteristics that can be used for tumor imaging, ability to be degraded in the TME, and the capacity to generate ROS via the Fenton-like reaction. In addition, Cu-based NMs can be used to inhibit tumor growth by carrying various types of drugs (DFS, DOX, or R837), not only for tumor-site-targeted delivery, but also for synergistic effects of various therapeutic modalities, such as PTT, PDT, CDT, and immunotherapy.

However, some important considerations need to be addressed. Firstly, there is no consensus on the best way to synthesize Cu-based NMs; the preparation and use of Cu-based NMs may generate wastewater and exhaust fumes, which could cause potential threats to environmental safety that should be extensively investigated and controlled. The most environmentally friendly way of synthesizing Cu-based NMs on a wide scale for clinical use is still being researched in order to minimize their detrimental impact on humans and the natural world. Cu-based NMs have not yet been used in actual clinical practice because the long-term biosafety, pharmacokinetics, and immunoreactivity of Cu-based NMs has not been adequately explored and only the short-term safety of Cu-based NMs has been validated in tumor-bearing mice. Therefore, their safety and toxicity must be completely established through long-term rigorous mammalian experimental testing. 

In conclusion, the application of Cu-based NMs in preclinical therapies has considerable advantages. Combining Cu-based NMs with PTT/PDT shows great promise for anti-tumor therapy. Our most critical mission is to promote human health to realize accurate, safe, and effective cancer treatments. Cu-based NMs produced for clinical application should be more precise and careful, but we still look forward to the development of more Cu-based NMs or other nanocomposites which have undergone comprehensive biosafety testing and have proven to be reliable.

## Figures and Tables

**Figure 1 pharmaceutics-15-02293-f001:**
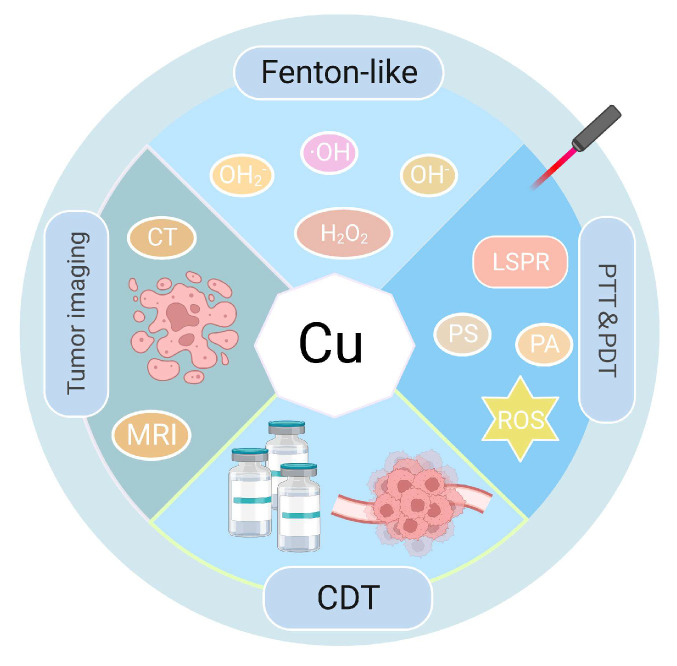
Functions of Cu and therapeutic involvement. CDT: chemodynamic therapy.

**Figure 2 pharmaceutics-15-02293-f002:**
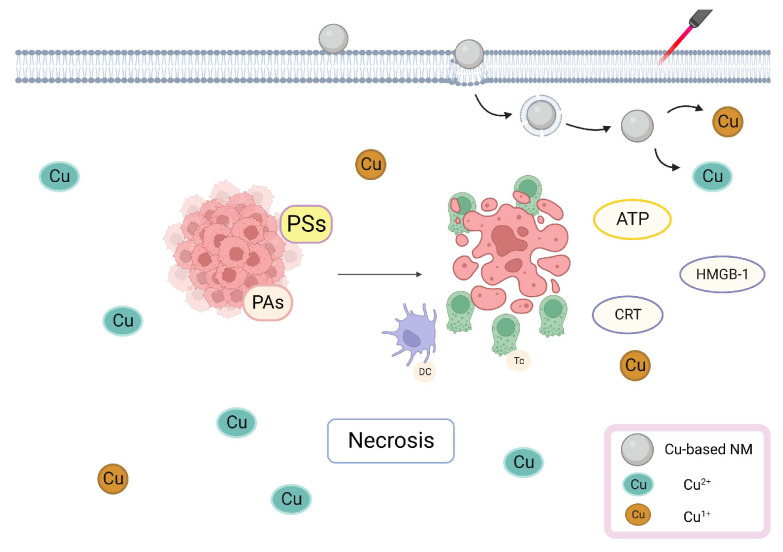
Cu-Based NMs synergizes PTT/PDT-induced tumor cell necrosis. Cu-containing PSs/PAs attached to tumor cells lead to tumor cell necrosis, stimulating the immune system by DAMPs.

**Figure 3 pharmaceutics-15-02293-f003:**
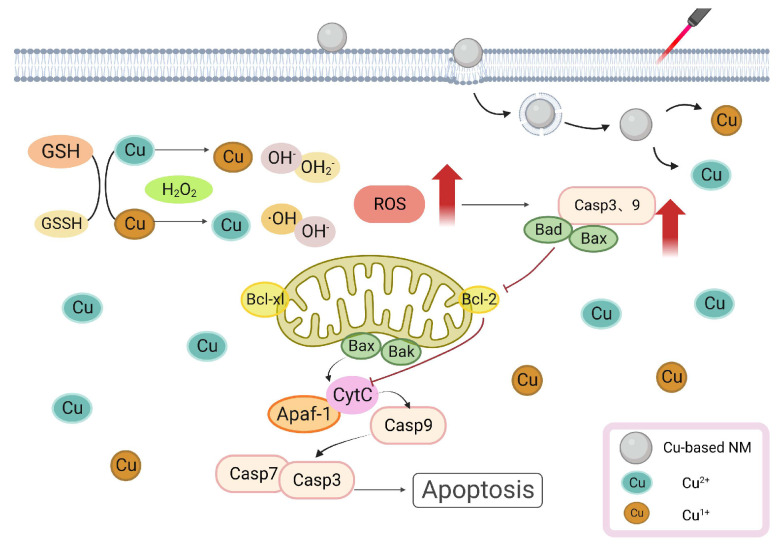
Cu-Based NMs synergizes PTT/PDT-induced apoptosis. Cu increased Bax, Bad, and caspases 3 and 9 levels, activated the caspase cascade response, and induced apoptosis.

**Figure 4 pharmaceutics-15-02293-f004:**
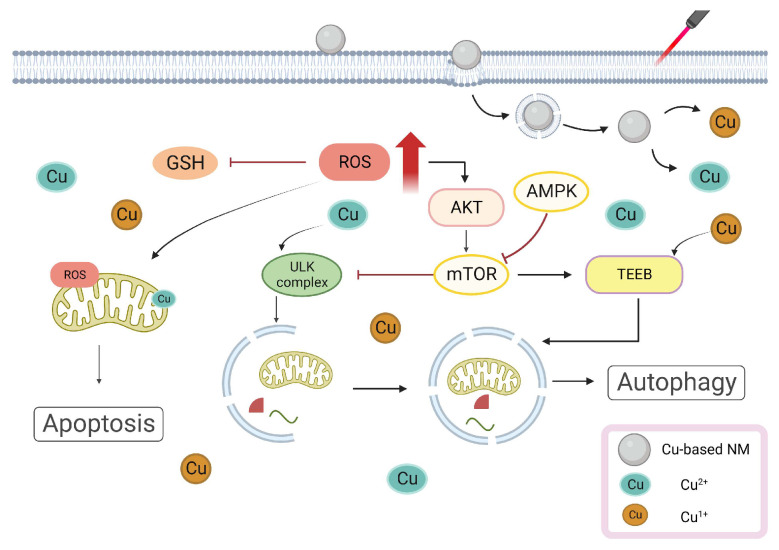
Cu-Based NMs synergizes PTT/PDT-induced autophagy. Autophagy is activated and the mTOR-dependent signaling is also interfered by the increased intracellular Cu concentration.

**Figure 5 pharmaceutics-15-02293-f005:**
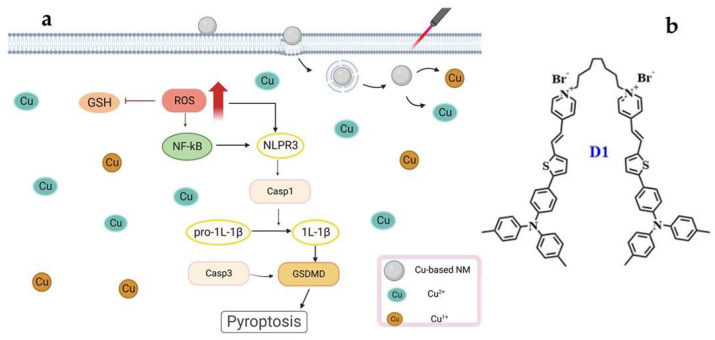
(**a**) Cu-Based NMs synergizes PTT/PDT-induced pyroptosis. Cu could activate the generation of ROS under irradiation, resulting in cancer cell pyroptosis. (**b**) The structure of dimeric photosensitizer D1.

**Figure 6 pharmaceutics-15-02293-f006:**
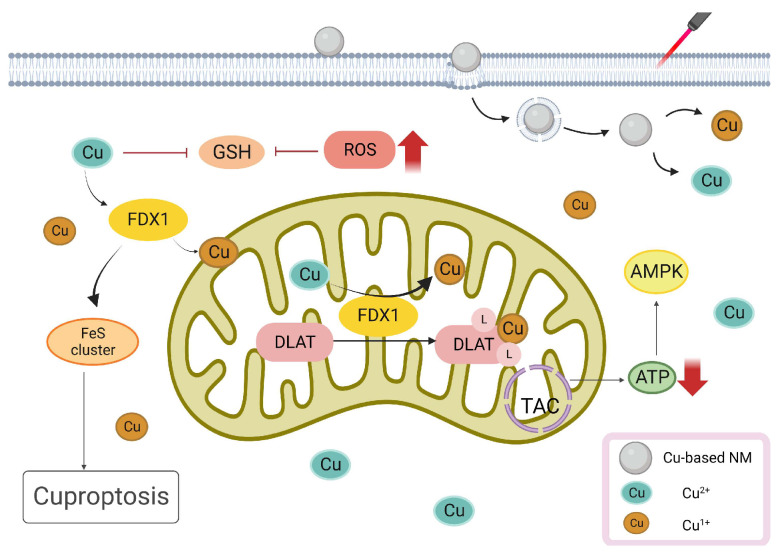
Cu-Based NMs synergizes PTT/PDT-induced cuproptosis. Increasing Cu levels in tumor cells can enhance NIR-responsive capability, cause cuproptosis, and synergize PTT/PDT tumor treatment.

**Figure 7 pharmaceutics-15-02293-f007:**
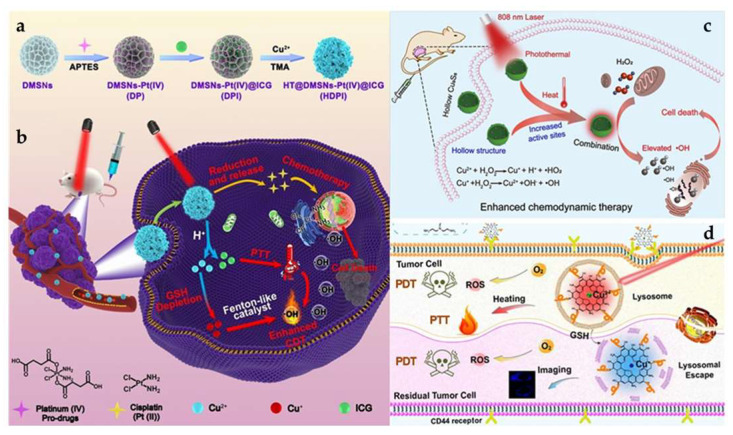
Anti-tumor therapeutic process of PTT/PDT with different kinds of Cu-based NMs. (**a**) Schematic representation of the synthesis process of a Cu-based NMs. And (**b**) [116] 2022 Elsevier, (**c**) [117] 2020 Elsevier and (**d**) [118] 2023 American Chemical Society are the schematic representation of the role of different Cu-based NMs in anti-tumor therapy.

**Figure 9 pharmaceutics-15-02293-f009:**
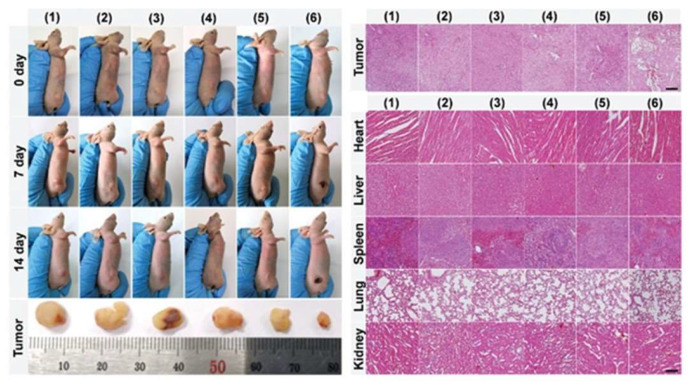
Tumor regression in Cu-based NMs applied in tumor-bearing mice and effects on vital organs. Scale bar: 100 μm. Ref. [169] 2021 John Wiley and Sons.

## Data Availability

Not applicable.

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
