# Peer review of "Recent Progress of Copper-Based Nanomaterials in Tumor-Targeted Photothermal Therapy/Photodynamic Therapy"

_pharmaceutics, 2023, doi:10.3390/pharmaceutics15092293_

Round 1

Reviewer 1 Report

the authors reviewed Cu-based NanoMaterials (NMs) and explored the role of Cu in synergizing Photothermal Therapy/ PhotoDynamicTherapy (PTT/PDT) for tumor cell death, also providing a comprehensive overview of
the Cu-based NMs utilized in PTT/PDT. The review is well organiozed and well written.

Mandatory change; In the introducition, I suggest to cite a recent paper on the biomedical application of Cu-based NMs: 

Al Kayal et al. Molecules, 28, 5981, https://doi.org/10.3390/molecules28165981

Mandatory change: In  addition, the not original picture should be mentioned the original sources with permission to publish them.

Reviewer 2 Report

In lines 100 to 108, the authors mention the tissue penetration depth of light, tumor non specificity, and hypoxia in the tumor microenvironment among the notable drawbacks in actual use for the applications of PTT and PDT. The authors then suggest that Cu-based NMs may be incorporated into PTT and PDT as carriers or modified PAs or PSs to fix these drawbacks. Surely the authors need to elaborate or clarify this “carriers or modified PAs or PSs”.

The authors mention magnetic resonance imaging using copper based nanomaterials without specifying which of these are diamagnetic and which are paramagnetic (and can be used for magnetic resonance imaging).

The authors have provided several cartoons to illustrate the biological reactions of copper ions (Cu1+ and Cu2+). The challenge with these cartoon is that they only show Cu ions, yet it is not the ions that induce PTT. Rather, it is the Cu-based NMs. So where are the Cu-based NMs in the cartoons?

Furthermore, Authors might consider putting the legend in a box to separate it from the intracellular biological reactions. This is illustrated in Figure 2.

In line 264, the authors refer to a dimeric photosensitizer D1 as though it is shown somewhere in the manuscript, yet it is not. Authors might consider showing the structure of the dimer.

The authors present most functionalized nanomaterials, e.g. MoS2-CuO@BSA/R837 using the nomenclature of the source documents without describing the various components, e.g. BSA in this case, BSA = bovine serum albumin.

There are many typographical errors. These are dealt with on the manuscript itself using the pop-up tool on highlighted text.

Reviewer 3 Report

In the present manuscript by Zhou et al. entitled "Advancements in Copper-Based Nanomaterials for Targeted Photothermal Therapy and Photodynamic Therapy in Tumor Treatment" present a comprehensive analysis of the advancements and utilization of copper-based nanomaterials (NMs) in the domain of phototherapy (PT) for the purpose of combating tumors. The authors have done satisfactory work, but I do have a few major comments; if worked out by the authors, it will enhance the quality of the article and be valuable for the readers.

Upon conducting a comprehensive review and meticulous analysis of the aforementioned article, it becomes evident that there are several pertinent questions and areas of inquiry that warrant further investigation, exploration, or deliberation. These inquiries and areas of interest have the potential to serve as valuable guiding principles for future research endeavors, scholarly exploration, or scholarly discourse.

1. The present inquiry seeks to elucidate the enduring ramifications and safety parameters associated with copper-based nanomaterials (NMs) employed in the context of photothermal therapy (PTT) and photodynamic therapy (PDT).

2. Could you kindly provide an in-depth analysis of the precise mechanisms by which copper augments the efficacy of photothermal therapy (PTT) and photodynamic therapy (PDT)?

3. What are the obstacles encountered in the process of transferring the utilization of copper-based nanomaterials (NMs) from controlled laboratory investigations to real-world clinical applications? In what manner can copper-based nanomaterials (Cu-based NMs) be effectively amalgamated with extant cancer treatment modalities in order to augment therapeutic outcomes?

What are the most suitable methodologies for the synthesis of copper-based nanomaterials (NMs) in order to ascertain their stability, biocompatibility, and therapeutic efficacy?Inquiry pertains to the customization of Cu-based nanomaterials (NMs) with the aim of selectively targeting distinct tumor types or cancer stages.

What are the environmental ramifications associated with the production and utilization of copper-based nanomaterials (NMs) in the context of medical applications?

In what ways can the progression of technological advancements contribute to the augmentation of imaging, delivery, and therapeutic functionalities associated with Copper-based Nanomaterials (Cu-based NMs)?

 The present study delves into the ethical considerations that arise from the utilization of nanotechnology and copper-based nanomaterials (NMs) in human subjects. What are the authors view about this and how they gonna rectify it.

There are minor grammatical and formatting like ''There is a lack of spacing between the reference numbers "[3–6]" and the word "of." but that can be corrected easily during the revision or proofreading stage.

There are minor grammatical and formatting like ''There is a lack of spacing between the reference numbers "[3–6]" and the word "of." but that can be corrected easily during the revision or proofreading stage.

Round 2

Reviewer 3 Report

The article can be accepted